

# Disclosing the native blueberry rhizosphere community in Portugal—an integrated metagenomic and isolation approach

Anicia Gomes[1,2], Rodrigo Narciso[1,2], Laura Regalado[1,2], Margarida Cardeano Pinheiro[1,2], Filipa Barros[1,2], Sara Sario[1,2], Conceição Santos[1,2] and Rafael J. Mendes[1,2]

[1] Department of Biology, Faculty of Sciences, University of Porto, Porto, Portugal
[2] LAQV-REQUIMTE, Department of Biology, Faculty of Sciences, University of Porto, Porto, Portugal

Corresponding author
Rafael J. Mendes,
rafael.mendes@fc.up.pt

## ABSTRACT

**Backgorund:** The production of red fruits, such as blueberry, has been threatened by several stressors from severe periods of drought, nutrient scarcity, phytopathogens, and costs with fertilization programs with adverse consequences. Thus, there is an urgent need to increase this crop's resilience whilst promoting sustainable agriculture. Plant growth-promoting microorganisms (PGPMs) constitute not only a solution to tackle water and nutrient deficits in soils, but also as a control against phytopathogens and as green compounds for agricultural practices.

**Methods:** In this study, a metagenomic approach of the local fungal and bacterial community of the rhizosphere of *Vaccinium corymbosum* plants was performed. At the same time, both epiphytic and endophytic microorganisms were isolated in order to disclose putative beneficial native organisms.

**Results:** Results showed a high relative abundance of *Archaeorhizomyces* and *Serendipita* genera in the ITS sequencing, and *Bradyrhizobium* genus in the 16S sequencing. Diversity analysis disclosed that the fungal community presented a higher inter-sample variability than the bacterial community, and beta-diversity analysis further corroborated this result. *Trichoderma* spp., *Bacillus* spp., and *Mucor moelleri* were isolated from the *V. corymbosum* plants.

**Discussion:** This work revealed a native microbial community capable of establishing mycorrhizal relationships, and with beneficial physiological traits for blueberry production. It was also possible to isolate several naturally-occurring microorganisms that are known to have plant growth-promoting activity and confer tolerance to hydric stress, a serious climate change threat. Future studies should be performed with these isolates to disclose their efficiency in conferring the needed resilience for this and several crops.

## INTRODUCTION

Blueberries (*Vaccinium* sp.) are one of the most important fruit crops in the world, due to their high levels of polyphenols, antioxidants, vitamins, minerals, and fibers (*de Souza et al., 2014*) that present numerous health benefits as anti-aging compounds, cancer prevention and of several degenerative diseases (*de Macêdo et al., 2017*; *Becker et al., 2019*). The high demand for healthier foods coupled with the increasing world population resulted in the expansion of the production of these fruits, with an average annual growth rate of 6.96% worldwide (*FAOSTAT, 2023*; *Knoema, 2023*). However, this crop is currently facing a number of challenges associated with climate change, as the global temperature is rising, contributing to longer and warmer summers, more and harsher heat waves, and a decline in rainfall levels, leading to severe drought periods and nutrient scarcity (*Lobos & Hancock, 2015*; *Linares et al., 2020*; *Smrke et al., 2021*). Besides climate issues, blueberry production finds several other obstacles, including high production costs associated with fertilization, weed management, and disease and pest control (*Yu et al., 2020*).

To survive under diverse threatening conditions, plants take advantage of the collaboration with their epiphytic and endophytic microbial communities (*Sarma et al., 2015*). These multipurpose beneficial communities are generally referred to as plant growth-promoting microorganisms (PGPMs) and support plants in water use, nutrient uptakes such as phosphorus and nitrogen, and the breaking down of minerals and organic matter (*Kumar, Patel & Meena, 2018*). They also compete for nutrients and produce antibiotics and lytic enzymes to prevent the development of pathogens (*Sattiraju et al., 2019*). On the other hand, plants secrete up to 40% of their photosynthesis products into the soil, supplying carbon to the microbiome population in the rhizosphere (*Berendsen, Pieterse & Bakker, 2012*).

A well-characterized group of PGPMs are fungi belonging to the *Trichoderma* genus. These microorganisms are a naturally occurring green strategy that plays a significant role in interacting with the plant's root system to boost growth and nutritional quality while producing secondary metabolites that influence pathogenic defense either directly as antibiotics or indirectly by eliciting the plant defense mechanisms against different pathogens (*Vinale & Sivasithamparam, 2020*). For instance, the up-regulation of defense-related pathways increases the expression of volatile organic compounds (VOCs), namely jasmonic acid and ethylene, that trigger induced systemic resistance, or salicylic acid, driving wound repair and systemic acquired resistance (*Ferreira & Musumeci, 2021*).

Currently, one of the most important challenges of environmental sustainability is to meet the needs of a rapidly growing human population; however, cultivable land is limited, so it is paramount to adopt appropriate agrobiotechnological measures to maximize food production while minimizing pollution and remediating contaminated soil (*Abhilash et al., 2016*). To this end, promoting native microorganisms' growth is essential to strengthening the plants' resilience to a multitude of abiotic and biotic challenges.

Nowadays, profiling microbial samples from the soil is drastically facilitated by the development of novel metagenomic tools that surpass traditional approaches, avoiding the ineffective culturing step. In fact, the cultivation and isolation of microorganisms provide
very limited results, as only around 1% of the soil microbiome is culturable, remaining the majority unexplored. Fortunately, metagenomics emerged as a driving force to survey the biodiversity of microbial communities (*Daniel, 2005*; *Wolińska, 2019*).

In Portugal, blueberry production has increased considerably in the last years, placing the country in the top 10 of the world's biggest producers, with an estimated annual growth of 31.27% (*FAOSTAT, 2023*; *Knoema, 2023*), significantly impacting Portuguese agriculture and economy (*Hilário et al., 2020*). However, to date, only a single study in Portugal has dwelled on the microbial communities associated with the rhizosphere of blueberry plants, being this study focused only on the bacterial community (*Gonçalves et al., 2022*).

Despite the previous existence of studies that seek to know the community of microorganisms in blueberry plants, it is of great necessity to know both bacterial and fungi communities present in a native form. In this way, this knowledge will be meaningful to disclose more specific strategies to improve local blueberry production, taking the most out of the indigenous nature attributes, and avoiding the need for agrochemicals that degrade soil quality and the desired fruit. Therefore, and in the frame of the European Green Deal, with special emphasis on the Farm to Fork strategy, this work aimed to characterize the local rhizosphere-colonizing microbiome from blueberry plants in a biological orchard in the Douro region of Portugal. In order to achieve this, we resort to metagenomic sequencing of bacterial 16S and fungal ITS regions, and at the same time, isolation and identification, through molecular characterization, of the native mycorrhiza rhizobium. This allowed the establishment of a first report on the total profiling of the native community of blueberries' rhizosphere in Portugal and the assessment of the putative presence of key native PGPMs.

## MATERIALS AND METHODS

### Sample selection and collection

For rhizosphere characterization, root, and soil samples from 6-year-old Blueberry plants, namely, *Vaccinium corymbosum* 'Brigitta' variety, grown under appropriate soil conditions, biological agricultural practices, and without the presence of other plant species, were collected in the Enxertada Indie Farmers production (Resende, 41°07′13.4″N 7°53′17.8″W) in October 2021. The 'Brigitta' variety was selected due to its high commercial value and belongs to the Northern highbush blueberry groups that are representative of the blueberry production in the north and center of Portugal, including in the sampling region. For root samples, roots with arbuscular morphology of *V. corymbosum* plants were selected; for soil samples, soil containing arbuscular roots with a depth of 5 cm was collected.

A total of 10 *V. corymbosum* plants were randomly chosen and one soil sample per plant was collected ($n = 10$) (R1 to R10), to characterize the representative bacterial and fungal microorganisms' populations at the site. The soil was placed in sterile bags and transported to the laboratory, and stored at −80 °C for further processing. For the root samples, another nine *V. corymbosum* plants were randomly chosen and the samples were collected in duplicate per plant ($n = 9$). The roots were also placed in sterile bags, and transported to

the laboratory to be processed within 24 h of collection, for isolation of epiphytic and endophytic fungi and bacteria.

Field sampling was approved by Enxertada Indie Farmers under the project STOP. SUZUKII.

### ITS and 16S amplicon metagenomic sequencing of *V. corymbosum* rhizosphere

Genomic DNA from soil for metagenomic analysis was obtained. For the ten soil samples, the Soil DNA Purification Kit (EURx®, Gdansk, Poland) was used following the manufacturer's instructions with a slight modification, namely, the Bead Tube containing 100 mg of sample was first snapfrozen in liquid nitrogen. DNA quality and quantity from every sample were carried out resorting to LVis Plate and the FLUOstar® Omega Multiplate reader (BMG, LABTECH, Ortenberg, Germany). To characterize the rhizobiome of the rhizosphere of *V. corymbosum* plants, amplicon metagenomic sequencing of the internal transcribed spacer (ITS) rRNA of the ITS1 region and 16S ribosomal RNA (rRNA) of the V3-V4 region was carried out on the DNA soil samples, at Novogene Europe (Novogene Co., Cambridge, UK). Briefly, after passing the quality control (QC), and amplification of the ITS and 16S regions, library construction was made with the amplicons (one library per sample). Sequencing by synthesis (SBS) technology was used resorting to the Illumina NovaSeq 6000 platform (Illumina®, San Diego, CA, USA) to generate 250 bp paired-end raw reads, and once sequencing was completed, QC and bioinformatics analysis were conducted.

### Metagenomic data analysis

For the metagenomic ITS and 16S amplicon sequencing analysis, the pipeline outsourced by Novogene was followed. After assessing the sequencing coverage and quality, Qiime (V1.7.0) and Uparse software (Uparse v7.0.1090) were used to identify the total tags obtained and operational taxonomic units (OTUs). Then, the top 10 highest abundant fungal and bacterial genus of *V. corymbosum* rhizosphere were obtained (R1 to R10), alongside the phylum relative abundance, resorting to blastall (Version 2.2.25) and Unite (V8.2) database. An evolutionary tree of the top 100 genera, and a phylogenetic tree of the top 10 genera identified in every sample were created using MUSCLE (Version 3.8.31). Also, alpha-diversity analysis for each sample was performed through diversity indexes (Shannon and Simpson) and richness indexes (Chao1 and ACE), and Venn diagrams to disclose the core microbiota, whilst, for beta-diversity analysis between the samples, a non-metric multidimensional scaling (NMDS) was applied. QIIME (Version 1.7.0) was used for both alpha and beta diversity analysis, and results were displayed with R software (Version 2.15.3; *R Core Team, 2013*).

### Isolation of endophytic and epiphytic microorganisms

To isolate epiphytic and endophytic fungi and bacteria, root samples were processed in two different ways. Healthy root segments were cut into various 3–5 cm segments. One segment was placed directly on potato dextrose agar (PDA) medium (27 g of potato

**Table 1 Primers used for ITS and 16S rRNA Sanger sequencing.**

| Primer | 5′ -> 3′ sequence | Reference |
|---|---|---|
| ITS | | |
| ITS1 | TCCGTAGGTGAACCTGCGG | *White et al. (1990)* |
| ITS4 | TCCTCCGCTTATTGATATGC | *White et al. (1990)* |
| 16S rRNA | | |
| 27F | AGAGTTTGATCCTGGCTCAG | *Lane (1991)* |
| 1492R | GGTTACCTTGTTACGACTT | *Lane (1991)* |

dextrose broth; 1.5% Agar; distilled water up to 1 L) supplemented with 50 mM kanamycin, to isolate epiphytic fungi, whilst for epiphytic bacteria, two different segments were placed directly on Luria Bertani (LB) medium (37 g of LB broth; 1.5% Agar; distilled water up to 1 L) and brain heart infusion (BHI) medium (20 g of BHI broth; 1.5% Agar; distilled water up to 1 L). For endophytic microorganisms, the segments were washed in sterile dH$_2$O and then went through a disinfection process consisting of 1-min washes each in 10% bleach, followed by 70% ethanol (EtOH) and 5 consecutive washes with sterile dH$_2$O. After drying, the root samples were also placed on PDA supplemented with 50 mM kanamycin, to isolate fungi, and for bacteria, two segments were placed on LB and BHI mediums, (one segment each). Plates were then incubated at 25 °C for 5–7 days and at room temperature for 2–3 days, for fungal and bacterial growth respectively. PDA medium was supplemented with an antibiotic to prevent the growth of bacteria during fungi isolation (*Vohník, 2020*). Morphologically different fungal and bacterial colonies were purified on new plates containing PDA medium and LB or BHI medium respectively, followed by incubation at 25 °C for 7 days and at room temperature for 2 days, respectively. In the end, a visual assessment was conducted to determine the morphological characteristics of the isolated fungi and bacteria. A photographic record was also performed. Fungal and bacterial isolates were stored in potato dextrose broth (PDB) and LB at −80 °C in 60% and 30% glycerol, respectively.

## Fungal and bacterial DNA extraction, amplification and sequencing

For genomic DNA extraction, the Plant & Fungi DNA Purification Kit (EURx®, Gdansk, Poland) was used for the 63 fungal isolates following the manufacturer's instruction, with 100 mg of fresh fungal tissue. For the 59 bacterial isolates, the Bacterial & Yeast Genomic DNA Purification Kit (EURx®, Gdansk, Poland) was used following the manufacturer's instructions. DNA quality and quantity from every bacterial and fungal sample were carried out resorting to LVis Plate and the FLUOstar® Omega Multiplate reader (BMG, LABTECH, Ortenberg, Germany). Endophytic and epiphytic fungal and bacterial isolates were identified by Sanger sequencing of the ITS and 16S partial region, respectively. To achieve that, a polymerase chain reaction (PCR) was carried out using a reaction mixture containing 1x Taq Master Mix (Bioron, Römerberg Germany), 1 µM of primers (listed in Table 1), and 1 ng of DNA template. The PCR cycle parameters were 2–5 min at 94 °C, followed by 25 cycles of 94 °C for 25 s, 49 °C or 55 °C (ITS, and 16S respectively) for

30 s, 72 °C for 50 s, and a final extension at 72 °C for 3 min. PCR amplicons were assessed by electrophoresis in a 1% agarose gel stained with GreenSafe Premium (NZYTech©, Lisboa, Portugal), at a constant voltage (120V) in 1x TBE buffer, and purified using the NZYGelpure Kit (NZYTech©, Lisboa, Portugal) following the manufacturer's instructions. Sequencing was outsourced to STAB Vida (Lisbon, Portugal).

### Culturable fungal and bacterial identification

To confirm the identity of the fungal and bacterial isolates, raw sequences of the Sanger sequencing of the ITS and 16S regions were assembled with Geneious Prime 2022.1 (Biomatters, Auckland, New Zealand), and then assessed by BLASTn for existing homology in the GenBank database (http://blast.ncbi.nlm.nih.gov/Blast.cgi). Isolates' identity was determined after assessing the query cover, E-value, and percentage of identity obtained.

## RESULTS

### Metagenomic data analysis

Illumina-sequencing of ITS amplicons produced 1,787,299 raw reads, that after qualification and removal of chimeras produced 1,293,940 effective tags, from which 556 OTUs were identified with an average length of 265 nt, and 97.98% Q20 score. Sequencing of 16S amplicons produced 1,800,579 raw reads, that after qualification and removal of chimeras produced 907 824 effective tags, from which 3,625 OTUs were identified with an average length of 413 nt and 98.24% Q20 score (Table 2). The fungal and bacterial communities associated with the rhizosphere of *V. corymbosum* from Enxertada Indie Farmers production are depicted in Figs. 1 and 2 (fungi), and Figs. 3 and 4 (bacteria) by an evolutionary tree of the 100 most abundant genera and top 10 most abundant genera in each sample. Among the fungal community, the Ascomycota (75.47%) and Basidiomycota (24.53%) were the most prevalently identified phyla (Fig. S1), while the bacterial community was dominated by Acidobacteriota (41.88%), Proteobacteria (29.48%) and Actinobacteria (15.06%) phyla (Fig. S2).

Among the top 100 most abundant genera of fungi in the analyzed samples, *Archaeorhizomyces* (31.49%), *Serendipita* (16.52%), *Atractospora* (15.26%), and *Penicillium* (7.57%) genera stood out as the most abundant genera among the total generated OTUs (Fig. 1). However, it is noticeable that *Serendipita* and *Penicillium* abundance was more homogenous across samples (Figs. 1 and 2). Despite less predominance among the whole fungal community, some particular genera dominated singular samples: *Galerina* (21.02% in R4) and *Pezoloma* (20.22% in R6) (Fig. 2).

The bacterial community composition of the collected samples disclosed more homogenous results as the 10 most abundant genera of bacteria represented only between 20.75% and 30.66% of the sample community (Fig. 3). Several genera were predominant across all samples, such as *Subgroup_2* (7.58%), *Bradyrhizobium* (7.32%), *Acidothermus* (6.29%), *Candidatus Solibacter* (5.11%), *Bryobacter* (4.72%) and *Acidibacter* (3.61%) (Fig. 4).
**Table 2 Sequencing data and quality of ITS and 16S rRNA sequencing.** Single, total and mean number of raw paired-end reads (raw reads), tags merged from reads (raw tags), cleaned and filtered tags (clean tags and effective tags) and operational taxonomic units (OTUs). Percentage of yielded effective tags in relation to produced raw reads. Single and mean number of sequenced effective tags nucleotides (base) and their average length. Single and mean percentage of Q20 and Q30 scores and GC content of effective tags.

| Soil sample | Raw reads | Raw tags | Clean tags | Effective tags | Effective tags (%) | OTUs | Base (nt) | Average length (nt) | Q20 (%) | Q30 (%) | GC content (%) |
|---|---|---|---|---|---|---|---|---|---|---|---|
| ITS |
| R1 | 155,590 | 134,367 | 121,460 | 120,129 | 77.21% | 425 | 32,349,810 | 269 | 98.6 | 96.56 | 55.01 |
| R2 | 184,863 | 162,014 | 122,527 | 121,987 | 65.99% | 282 | 36,837,738 | 302 | 96.97 | 92.77 | 54.85 |
| R3 | 171,154 | 155,678 | 131,205 | 130,046 | 75.98% | 215 | 39,584,999 | 304 | 96.62 | 92.04 | 54.51 |
| R4 | 182,557 | 157,156 | 109,792 | 109,178 | 59.80% | 417 | 32,475,292 | 297 | 97.52 | 93.89 | 51.19 |
| R5 | 170,089 | 161,707 | 149,993 | 149,260 | 87.75% | 330 | 42,733,120 | 286 | 99.15 | 97.32 | 54.99 |
| R6 | 146,530 | 110,773 | 92,765 | 90,347 | 61.66% | 598 | 23,273,395 | 258 | 98.22 | 96.05 | 50.33 |
| R7 | 194,018 | 134,255 | 121,547 | 115,467 | 59.51% | 1,217 | 29,139,615 | 252 | 99.07 | 97.49 | 53.93 |
| R8 | 185,367 | 142,641 | 142,450 | 140,579 | 75.84% | 979 | 32,195,048 | 229 | 97.54 | 95.85 | 50.62 |
| R9 | 197,154 | 157,997 | 157,911 | 157,324 | 79.80% | 435 | 35,314,304 | 224 | 97.82 | 95.21 | 51.12 |
| R10 | 199,977 | 164,373 | 160,783 | 159,623 | 79.82% | 657 | 37,195,561 | 233 | 98.29 | 96.42 | 52.92 |
| Mean | 178,730 | 148,096 | 131,043 | 129,394 | 72.34% | 556 | 34,109,888 | 265 | 97.98 | 95.36 | 52.95 |
| Total | 1,787,299 | 1,480,961 | 1,310,433 | 1,293,940 | | 5,555 | | | | | |
| 16S rRNA |
| R1 | 170,688 | 159,243 | 156,519 | 86,747 | 50.82% | 2,929 | 35,846,478 | 413 | 98.36 | 94.59 | 56.60 |
| R2 | 183,187 | 168,332 | 165,057 | 93,749 | 51.18% | 3,681 | 38,746,901 | 413 | 98.27 | 94.43 | 56.91 |
| R3 | 188,628 | 172,749 | 169,282 | 99,543 | 52.77% | 3,523 | 41,086,071 | 413 | 98.24 | 94.43 | 57.29 |
| R4 | 181,437 | 167,544 | 164,431 | 90,597 | 49.93% | 3,830 | 37,408,294 | 413 | 98.25 | 94.43 | 57.00 |
| R5 | 172,924 | 160,906 | 158,029 | 84,100 | 48.63% | 3,202 | 34,730,044 | 413 | 98.28 | 94.44 | 56.28 |
| R6 | 187,731 | 173,025 | 169,585 | 96,238 | 51.26% | 3,687 | 39,763,304 | 413 | 98.22 | 94.32 | 57.08 |
| R7 | 174,883 | 160,798 | 157,893 | 83,548 | 47.77% | 3,814 | 34,532,604 | 413 | 98.23 | 94.24 | 57.05 |
| R8 | 178,887 | 164,797 | 161,674 | 90,725 | 50.72% | 3,864 | 37,459,285 | 413 | 98.22 | 94.29 | 57.10 |
| R9 | 173,178 | 157,647 | 154,428 | 89,798 | 51.85% | 4,024 | 37,162,831 | 414 | 98.11 | 93.99 | 57.08 |
| R10 | 189,036 | 173,832 | 170,324 | 92,779 | 49.08% | 3,699 | 38,309,489 | 413 | 98.17 | 94.17 | 57.12 |
| Mean | 180,058 | 165,887 | 162,722 | 90,782 | 50.40% | 3,625 | 34,378,830 | 413 | 98.24 | 94.33 | 56.95 |
| Total | 1,800,579 | 1,658,873 | 1,627,222 | 907,824 | | 36,253 | | | | | |

## Alpha-diversity and beta-diversity of microbial communities

The richness (Chao1 and ACE indices) and diversity (Shannon and Simpson indices) of the communities were evaluated. As disclosed by the relative abundances of genera, analyzed samples showed a heterogenous diversity and richness of the fungal community, with Shannon and Simpson indices revealing a 30.19% and 23.81% coefficient of variation, while Chao1 and ACE indices displayed a 58.67% and 57.35% coefficient of variation (Table 3). Meanwhile, the bacterial communities disclosed very little variation of diversity and richness between samples, with the Shannon and Simpson indices presenting a 3.02% and 0.16% coefficient of variation whereas Chao1 and ACE indices presented a 9.65% and 9.40% coefficient of variation (Table 3).
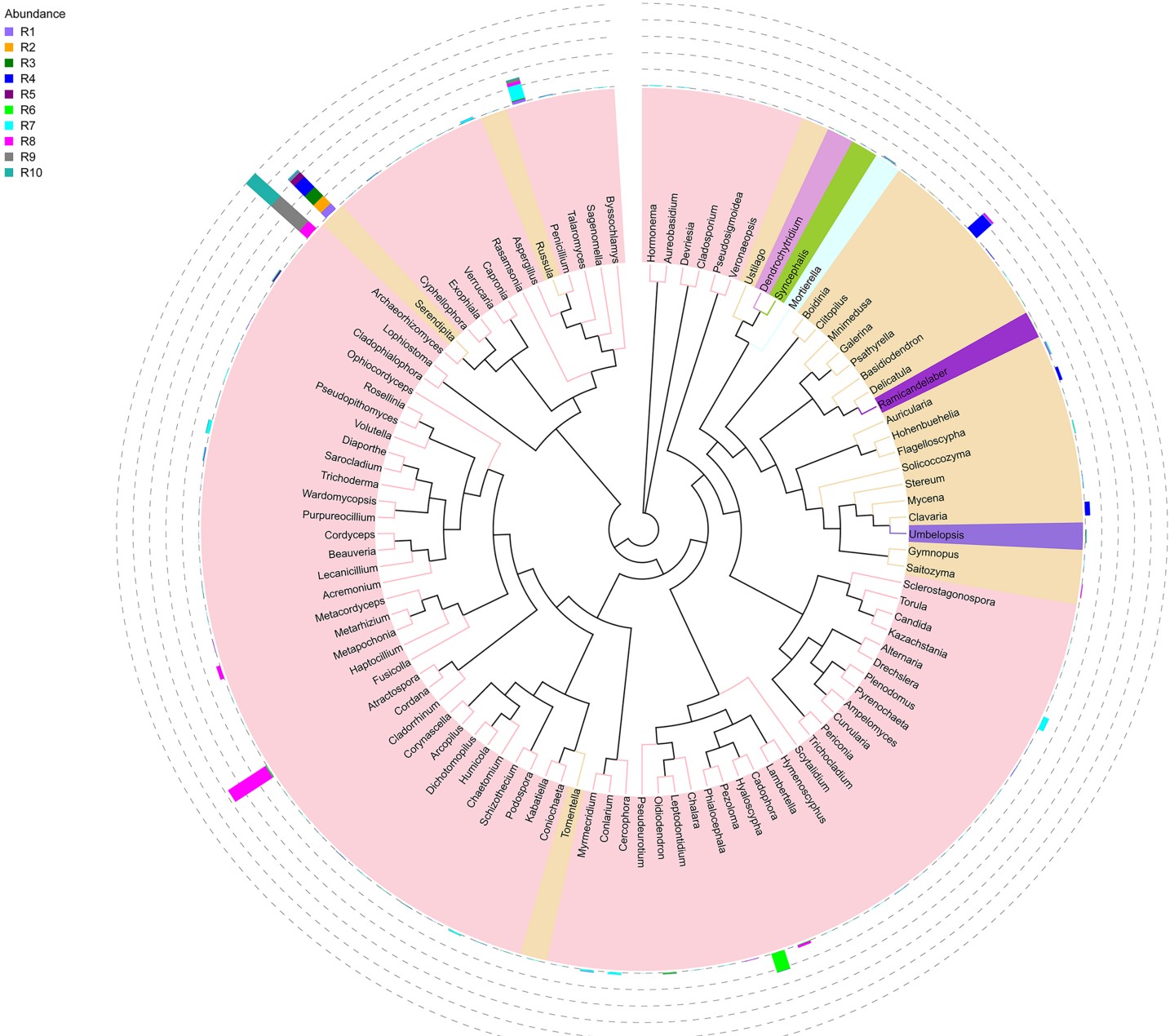

**Figure 1 Evolutionary tree of the 100 most abundant genera of fungi across the different samples.** Each bar represents the relative abundance of the genus in each sample. Each color of the branches corresponds to one phylum.

To further complement the analysis of inter-sample diversity, a multivariate analysis by non-linear multidimensional scaling (NMDS) was performed. The results showed a general dissimilarity between all fungal communities (Fig. 5), disclosing no correlation between any sample. On the other hand, the bacterial communities associated were shown to be generally more closely related to each other, especially R6, R7, R8, R9, and R10 samples, which grouped in the upper-right quadrant (Fig. 6).
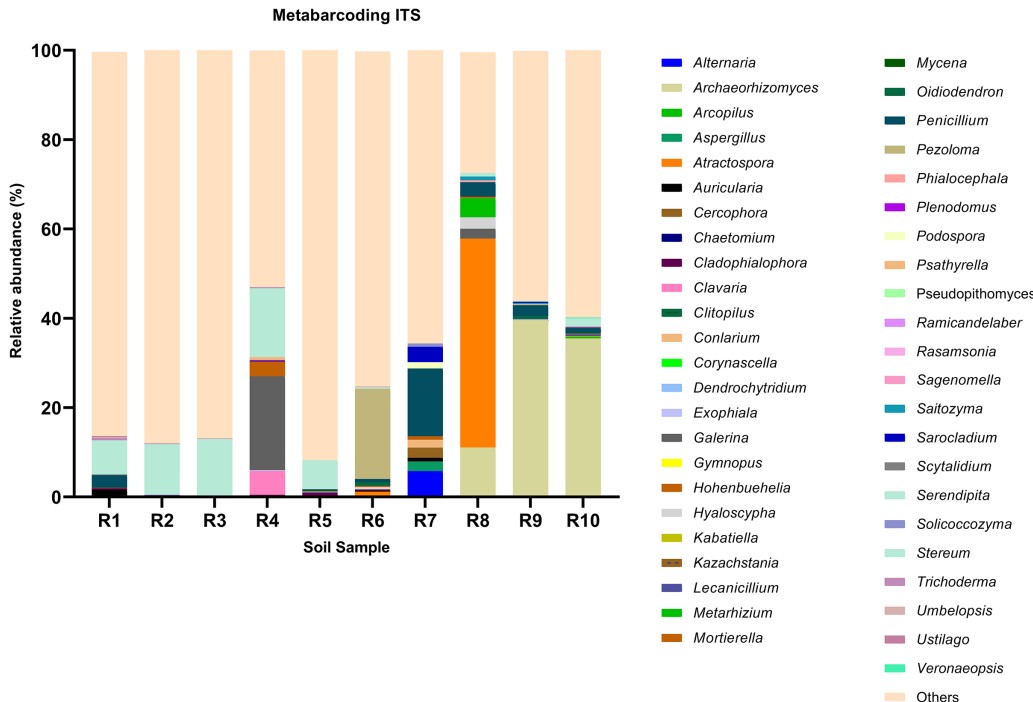

**Figure 2** Relative abundance of the top 10 most abundant genera identified by ITS sequencing in each sample.

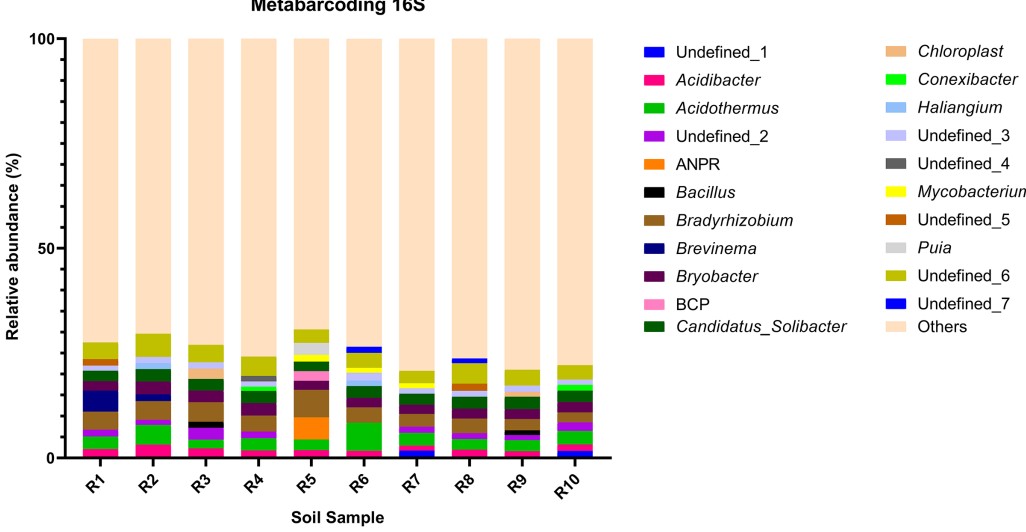

**Figure 3** Relative abundance of the top 10 most abundant genera identified by 16S sequencing in each sample.

## Core microbiota of rhizosphere samples

The alpha diversity of the fungal and bacterial communities was also analyzed regarding the common OTUs between samples. Although unique OTUs were found in every sample, there were 20 common OTUs on fungi identification (Fig. 7A). *Trichoderma* and *Serendipita* genera were the most abundant genera among the core fungi with three OTUs

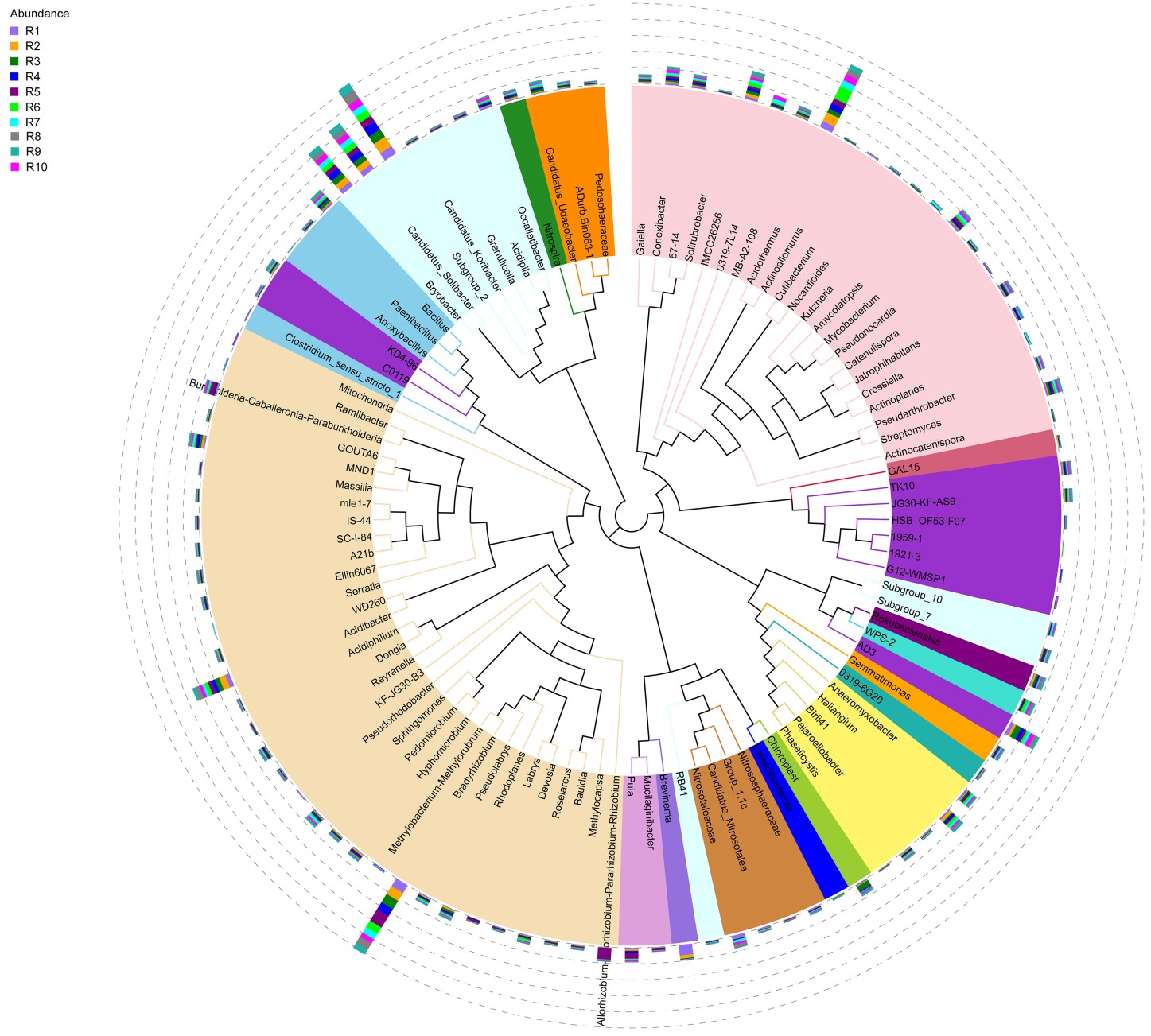

**Figure 4 Evolutionary tree of the 100 most abundant genera of fungi across the different samples.** Each bar represents the relative abundance of the genus in each sample. Each color of the branches corresponds to one phylum.

each (Table S1). Moreover, the bacterial community revealed a set of 1,152 common OTUs (Fig. 7B). The genera *Bryobacter* (27 OTUs), *Subgroup_2* (26 OTUs), *IMCC26256* (23 OTUs), *Candidatus_Solibacter* (21 OTUs), *MB-A2-108* (21 OTUs), *Haliangium* (19 OTUs), *67-14* (17 OTUs) and *Acidothermus* (17 OTUs), were frequently encountered among this set (Table S2).

**Table 3 Alpha diversity indexes of the fungal and bacterial communities of each sample.** Diversity indexes (Shannon and Simpson) and richness indexes (Chao1 and ACE) with respective mean, standard deviation (Stand. Dev.) and coefficient of variation (Coeff. Variation).

| Soil sample | Observed species | Shannon | Simpson | Chao1 | ACE |
|---|---|---|---|---|---|
| ITS | | | | | |
| R1 | 371 | 3.28 | 0.73 | 438.65 | 461.98 |
| R2 | 265 | 3.20 | 0.76 | 282.22 | 282.48 |
| R3 | 180 | 2.82 | 0.69 | 206.86 | 225.23 |
| R4 | 362 | 3.38 | 0.82 | 403.89 | 405.32 |
| R5 | 279 | 1.32 | 0.28 | 337.16 | 349.08 |
| R6 | 598 | 4.06 | 0.87 | 832.18 | 737.55 |
| R7 | 1,161 | 5.32 | 0.88 | 1,247.03 | 1,274.72 |
| R8 | 910 | 4.49 | 0.78 | 1,167.67 | 1,138.07 |
| R9 | 374 | 3.38 | 0.84 | 473.32 | 494.28 |
| R10 | 579 | 4.07 | 0.85 | 713.27 | 757.82 |
| **Mean** | | 3.55 | 0.74 | 625.44 | 625.80 |
| **Stand. Dev.** | | 1.07 | 0.18 | 366.93 | 358.91 |
| **Coeff. Variation** | | 30.19% | 23.81% | 58.67% | 57.35% |
| 16S | | | | | |
| R1 | 2,698 | 9.07 | 0.99 | 2,878.98 | 2,917.65 |
| R2 | 3,365 | 9.29 | 0.99 | 3,723.17 | 3,746.40 |
| R3 | 3,168 | 9.19 | 0.99 | 3,516.07 | 3,573.15 |
| R4 | 3,534 | 9.64 | 1.00 | 3,861.02 | 3,910.73 |
| R5 | 2,913 | 9.08 | 0.99 | 3,169.96 | 3,215.31 |
| R6 | 3,342 | 9.48 | 1.00 | 3,606.90 | 3,685.56 |
| R7 | 3,489 | 9.74 | 1.00 | 3,865.35 | 3,793.94 |
| R8 | 3,549 | 9.65 | 1.00 | 3,818.78 | 3,902.55 |
| R9 | 3,698 | 9.79 | 1.00 | 3,987.78 | 4,017.09 |
| R10 | 3,397 | 9.75 | 1.00 | 3,664.03 | 3,726.39 |
| **Mean** | | 9.43 | 0.99 | 3,567.14 | 3,607.96 |
| **Stand. Dev.** | | 0.28 | 0 | 344.34 | 339.09 |
| **Coeff. Variation** | | 3.02% | 0.16% | 9.65% | 9.40% |

## Identification of isolated fungal and bacterial communities from the rhizosphere

The isolation of endophytic and epiphytic microorganisms yielded a total of 167 fungal and 288 bacterial isolates from the nine root samples collected. Of this total, after assessing each one for different morphological characteristics, isolates were grouped together resulting in 63 fungi and 59 bacteria isolates (Tables S3 and S4). From these, 20 endophytic and 18 epiphytic unique fungi and nine endophytic and 18 epiphytic unique bacteria were identified by BLASTn (Table 4). A list of identification results of endophytic and epiphytic fungi and bacteria isolates with their E-value, query cover, and identity were summarized in Tables S1 and S2.

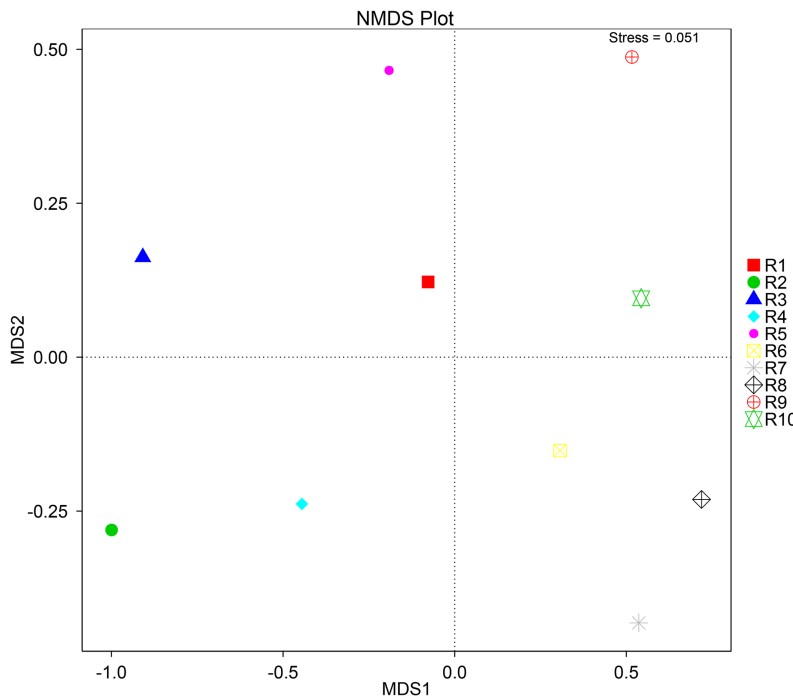

**Figure 5 Multivariate analysis of the fungal communities of each soil sample by non-linear multidimensional scaling (NMDS) to evaluate the dissimilarity between each associated community.**

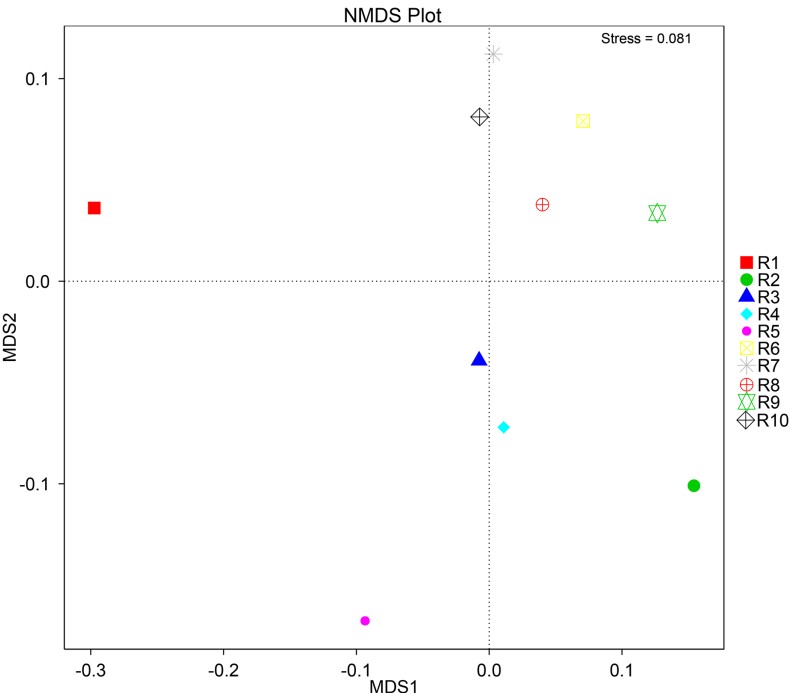

**Figure 6 Multivariate analysis of the bacterial communities of each soil sample by non-linear multidimensional scaling (NMDS) to evaluate the dissimilarity between each associated community.**

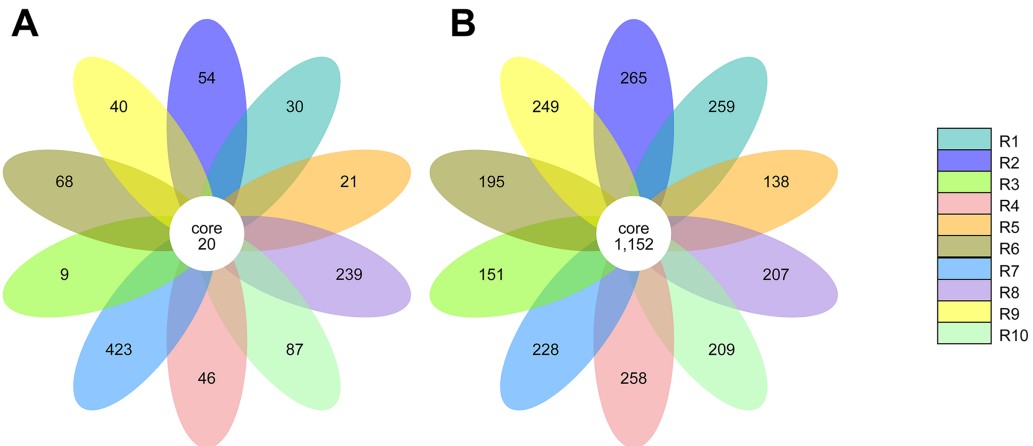

**Figure 7 Venn diagram of the common operational taxonomic units (OTUs) of the ITS (A) and 16S (B) metabarcoding.** Each petal in the flower diagram represents for a sample or group, with different colors for different samples. The core number in the center is for the number of OTUs present in all samples, while number in the petals for the unique OTUs only showing in each sample.

Regarding the fungal community, different species of the *Trichoderma* genus were identified (*Trichoderma asperellum*, *T. atroviride*, *T. citrinoviride*, *T. crissum*, *T. gamsii*, *T. hamatum*, *T. harzianum*, *T. koningii*, *T. longibrachiatum*, *T. spirale*, *T. sulphureum*, *T. virens*,), both endophytically and epiphytically. Furthermore, other genera were identified: *Cunninghamella*, *Diaporthe*, *Fusarium*, *Macrophomina*, *Mucor*, *Oidiodendron*, *Penicillium*, *Phomopsis*, *Rosellinia*, *Sclerotium*, *Setophoma* and *Trametes*. Several bacterial genera were identified, comprising *Acinetobacter*, *Bacillus*, *Chryseobacterium*, *Citrobacter*, *Enterobacter*, *Erwinia*, *Klebsiella*, *Leclercia*, *Leifsonia*, *Lelliottia*, *Lysinibacillus*, *Micrococcus*, *Paenibacillus*, *Pantoea*, *Priestia*, *Pseudomonas*, *Staphylococcus* and *Stenotrophomonas*.

Among the identified genera, *Trichoderma* (49.20%), *Fusarium* (14.28%), and *Phomopsis* (11.1%) were the most prevalent fungi, followed by *Diaporthe* (6.34%), *Mucor* (4.76%) and *Penicillium* (3.17%), whilst for bacteria, the most prevalent were *Bacillus* (27.12%), followed by *Priestia* (16.95%), *Lysinibacillus* (13.56%), *Pseudomonas* (10.17%), *Pantoea* (5.08%), *Paenibacillus* (3.39%), *Micrococcus* (3.39%), and *Klebsiella* (3.39%).

## DISCUSSION

Blueberry production has been facing several challenges in the form of drought and nutrient scarcity due to the ongoing climate change, the unregulated application of chemical fertilizers, and diseases caused by several pathogens (*Tournas & Katsoudas, 2005*; *Guo et al., 2021a*; *Jiang et al., 2022*; *Zhang, Liu & Zhang, 2022*; *Zhao et al., 2022*). Thus, an urgent strategy to tackle these issues, whilst following the call for sustainable agriculture as proposed by the European Green Deal is in serious need. To answer this call, several microorganisms, from plant growth-promoting bacteria to antagonistic microorganisms and mycorrhizal fungi have been extensively explored as a solution (*Redman et al., 2021*; *Acuña-Rodríguez et al., 2022*; *Bello et al., 2022*; *Wei et al., 2022*). With this in mind, the characterization of the native fungal and bacterial community structure of the rhizosphere

**Table 4 List of isolated endophytic and epiphytic fungi and bacteria.**

| | Code | Type | Species | | Code | Type | Species |
|---|---|---|---|---|---|---|---|
| ITS | F1 | Endophytic | *Diaporthe columnaris* | 16S rRNA | B1 | Endophytic | *Bacillus* sp. |
| | F2 | | *Diaporthe* sp. | | B2 | | *Enterobacter* sp. |
| | F3 | | *Fusarium diaminii* | | B3 | | *Erwinia billingiae* |
| | F4 | | *Fusarium oxysporum* | | B4 | | *Lysinibacillus* sp. |
| | F5 | | *Fusarium* sp. | | B5 | | *Micrococcus* sp. |
| | F6 | | *Mucor moelleri* | | B6 | | *Paenibacillus* sp. |
| | F7 | | *Penicillium adametzii* | | B7 | | *Pantoea* sp. |
| | F8 | | *Phomopsis* sp. | | B8 | | *Priestia* sp. |
| | F9 | | *Rosellinia necatrix* | | B9 | | *Pseudomonas* sp. |
| | F10 | | *Sclerotium glucanicum* | | B10 | Epiphytic | *Acinetobacter* sp. |
| | F11 | | *Setophoma terrestris* | | B11 | | *Bacillus thurigiensis* |
| | F12 | | *Trametes villosa* | | B12 | | *Bacillus* sp. |
| | F13 | | *Trichoderma atroviride* | | B13 | | *Chryseobacterium* sp. |
| | F14 | | *Trichoderma hamatum* | | B14 | | *Citrobacter* sp. |
| | F15 | | *Trichoderma harzianum* | | B15 | | *Klebsiella* sp. |
| | F16 | | *Trichoderma koningii* | | B16 | | *Leclercia* sp. |
| | F17 | | *Trichoderma longibrachiatum* | | B17 | | *Leifsonia* sp. |
| | F18 | | *Trichoderma spirale* | | B18 | | *Lelliottia amnigena* |
| | F19 | | *Trichoderma virens* | | B19 | | *Lysinibacillus* sp. |
| | F20 | | *Trichoderma* sp. | | B20 | | *Micrococcus* sp. |
| | F21 | Epiphytic | *Cunninghamella elegans* | | B21 | | *Paenibacillus* sp. |
| | F22 | | *Diaporthe columnaris* | | B22 | | *Pantoea* sp. |
| | F23 | | *Fusarium oxysporum* | | B23 | | *Priestia megaterium* |
| | F24 | | *Macrophomina phaseolina* | | B24 | | *Priestia* sp. |
| | F25 | | *Mucor moelleri* | | B25 | | *Pseudomonas* sp. |
| | F26 | | *Oidiodendron maius* | | B26 | | *Staphylococcus* sp. |
| | F27 | | *Penicillium paraherquei* | | B27 | | *Stenotrophomonas* sp. |
| | F28 | | *Phomopsis columnaris* | | | | |
| | F29 | | *Phomopsis* sp. | | | | |
| | F30 | | *Trichoderma asperellum* | | | | |
| | F31 | | *Trichoderma atroviride* | | | | |
| | F32 | | *Trichoderma citrinoviride* | | | | |
| | F33 | | *Trichoderma crassum* | | | | |
| | F34 | | *Trichoderma gamsii* | | | | |
| | F35 | | *Trichoderma hamatum* | | | | |
| | F36 | | *Trichoderma spirale* | | | | |
| | F37 | | *Trichoderma sulphureum* | | | | |
| | F38 | | *Trichoderma* sp. | | | | |

of *V. corymbosum* 'Brigitta' could disclose several microorganisms that could be explored as agents to improve the resilience of the plants, promote plant growth and development, and pose as a second key player against plant diseases.

In this study, the Ascomycota and Basidiomycota phyla expectedly dominated the fungal communities. There have been previous reports of the high predominance of these groups associated with the rhizosphere of not only *Vaccinium* spp. (*Li et al., 2020*; *Dong et al., 2022*; *Zhou et al., 2022*), but other crop and non-crop plants as well (*Fuentes et al., 2020*; *Cheng et al., 2022*; *Liu et al., 2022*). This is easily explained as these phyla encompass a plethora of plant-associated microbes, such as mycorrhizal fungi, saprophytic organisms, and plant pathogens (*Berbee, 2001*; *Hannula et al., 2012*). *Serendipita* and *Peniccilium* were two of the most abundant genera distributed across all or most of the collected samples. *Serendipita* spp., such as *Serendipita indica* or *S. vermifera*, have been previously described as mycorrhizal endophytic fungi of several plants, which confer protection towards infection, detoxification of xenobiotics, tolerance to hydric and mechanical stress, enhancement of plant and root growth and modification of soil physicochemical properties (*Sun et al., 2014*; *Hosseini, Mosaddeghi & Dexter, 2017*; *Sarkar et al., 2019*; *Hosseini & Mosaddeghi, 2021*; *Wang et al., 2022*). Moreover, *Trzewik, Marasek-Ciolakowska & Orlikowska (2020)* reported protective behavior of *S. indica* towards infection by the phytopathogen *Phytophthora cinnamomic* in *V. corymbosum*, extending the beneficial interaction of *Serendipita* spp. with blueberry plants (*Trzewik, Marasek-Ciolakowska & Orlikowska, 2020*). On the other hand, *Penicillium* spp. are common soil colonizers due to their key role in phosphorous cycling, an important macronutrient, as they are able to solubilize it for plant uptake (*Wakelin et al., 2004*; *Pandey et al., 2008*; *Sharma et al., 2013*; *de Oliveira Mendes et al., 2014*). In fact, the application of *Penicillium* spp., as biofertilizers is already commercially available (*Richardson et al., 2009*). The particular association of this genus with *V. corymbosum* has also been reported and co-inoculation of mycorrhizal fungi and *Penicillium chrysosporium* has been shown to promote the growth of blueberry plants in early stages (*Arriagada et al., 2012*; *Pescie et al., 2021*). Other genera were found to be prominent in singular or few of the collected samples, namely the *Archaeorhizomyces*, *Atractospora*, and *Pezoloma* genera. Despite scarce information about the *Archaeorhizomyces* genus, it is known to ubiquitously colonize soil samples (*Rosling et al., 2011*). However, the nature of its relationship with the plant and other mycorrhizal fungi still remains unknown (*Choma et al., 2016*; *Cruz-Paredes et al., 2019*). Nevertheless, association with *Vaccinium* spp. has been previously reported (*Li et al., 2020*; *Rodriguez-Mena et al., 2022*). The *Atractospora* genus is mainly characterized as saprophytic, associated with submerged wood in freshwater (*Réblová, Fournier & Štěpánek, 2016*). *Atractospora* spp. have been previously identified in different soil samples (*Xie et al., 2019a*; *Shen et al., 2020*; *Lemmel et al., 2021*), yet plant-fungus relationships have not been dwelled on. Ericoid mycorrhizal fungi (EMF) interact closely with Ericaceae plants, in which *Vaccinium* spp. are included, establishing mutualistic relationships characterized by the exchange of carbon, nitrogen, and phosphorous compounds (*Read, Leake & Perez-Moreno, 2011*; *Leopold, 2016*). *Pezoloma* is a known genus of EMF and was the most abundant genus in the R6 sample. Although this

was not a ubiquitously present genus in our samples, it has been previously described as part of the core rhizobiome of different *Vaccinium* spp. (*Li et al., 2020*). Additionally, other EMF have been identified in our samples, namely, the genera *Oidiodendron*, which has been previously associated with conferring zinc tolerance (*Vallino et al., 2009*; *Khouja et al., 2013*; *Chiapello, Martino & Perotto, 2015*) and *Hyaloscypha* (*Leopold, 2016*; *Fehrer et al., 2019*; *Wei & Chen, 2022*). The *Alternaria* genus was also part of the rhizobiome of several of the analyzed samples. This genus is composed of both saprophytic and pathogenic species by producing host-selective toxins responsible for tissue necrosis (*Thomma, 2003*). *Alternaria atenuata* and *A. tenuissima* have already been reported to have caused leaf spot diseases in *V. corymbosum* in different countries (*Kwon et al., 2014*; *You et al., 2014*; *Nadziakiewicz et al., 2018*). Identification of phytopathogenic microorganisms is of great importance to better establish control plans (*Hanson et al., 2000*). Gathering these observations, we conclude that the identified genera that compose the rhizobiome structure of the collected samples were overall expected as they compose both ubiquitous and mutualistic organisms of *V. corymbosum*.

Moreover, the predominance of the Acidobacteria, Proteobacteria, and Actinobacteria phyla regarding the bacterial communities also comes with no surprise as previous reports have stated the abundance of these groups, not only in rhizosphere soils of several plants (*Fuentes et al., 2020*; *Cheng et al., 2022*), but in *Vaccinium* spp. in particular (*Li et al., 2020*; *Gonçalves et al., 2022*; *Rodriguez-Mena et al., 2022*). The presence of Acidobacteria seems to be dependent on the soil pH conditions and, in general, acidic soils seem to favor the growth of these bacteria (*Sait, Davis & Janssen, 2006*; *Rousk et al., 2010*; *Darnell & Cruz-Huerta, 2011*). However, this correlation is also dependent on the subgroups of this phylum (*Sait, Davis & Janssen, 2006*; *Rousk et al., 2010*). In our study, *Bryobacter*, *Subgroup_2*, and *Candidatus Solibacter* were three of the most dominant genera regarding the overall bacterial composition of our samples, belonging to subgroup 1, 2, and 3 of Acidobacteria. This is in accordance with the required acidity to grow blueberry plants, which has been proposed to not surpass pH 6 as it causes nutrient uptake deficiency (*Darnell & Cruz-Huerta, 2011*). Moreover, some species of several subgroups of Acidobacteria have been shown to play an important role in nutrient-deficient soils and inhospitable habitats (*Rawat et al., 2012*). *Bradyrhizobium* and *Acidibacter* were the most dominant genera in the Proteobacteria phylum. Inoculation of *Bradyrhizobium* spp. has been proven to have a positive impact on the growth of some legumes due to their nitrogen-fixing capacity (*Elsheikh & Ibrahim, 1999*; *Egamberdiyeva, Qarshieva & Davranov, 2004*; *Bedmar, Robles & Delgado, 2005*; *Badawi, Biomy & Desoky, 2011*; *Delamuta et al., 2015*). Nonetheless, this is not the first time this genus was identified as one of the most abundant genera associated with the rhizosphere of *Vaccinium* spp. and, simultaneously, with higher nitrogen content in blueberry plants (*Morvan et al., 2020*). The *Acidibacter* genus, similar to the Actinobacteria genus *Acidothermus* has been already reported as one of the most abundant genera of *Vaccinium* spp. rhizobiome (*Chen et al., 2019*; *Guo et al., 2021b*; *Rodriguez-Mena et al., 2022*; *Wang, Sun & Xu, 2022*). The relevance of these genera is stressed in previous reports that prove that their relative abundance has been shown to be modulated according to the soil nutrient availability,

namely upon application of fertilizers, suggesting this group is an indicator of soil properties (*Fu et al., 2021*; *Ren et al., 2021*). Our results reveal the presence of bacterial taxa common in blueberry plants' rhizosphere, simultaneously reflecting the physicochemical properties of the soil, which are important modulators of bacterial ecology (*Li et al., 2017*; *Tiquia et al., 2002*; *Zhang et al., 2017*).

Despite showing expected fungal and bacterial taxa, our analysis revealed an increased heterogeneity among the fungal communities across the different samples. The alpha diversity analysis proves that both diversity and richness indices disclose a higher variation than the bacterial communities. Corroborating these data, the beta diversity analysis by NMDS clustered the samples according to the 16S sequencing, while no correlation between samples was found according to the ITS sequencing. This observation may be justified by a higher sensitivity by fungi upon soil properties' shifts, namely water, nitrogen, and carbon content (*He et al., 2017*; *Kaisermann et al., 2015*; *Wang et al., 2019*). Furthermore, fungi and plant, namely arbuscular mycorrhizal fungi, establish spatial, physiological, and plant attributes-dependent structures, which contribute to variation in the identified community even in samples from the same root system (*Deveautour et al., 2021*; *Mummey & Rillig, 2008*; *Powell & Bennett, 2016*).

The core rhizobiome was another parameter of great importance as it aided in a better characterization of the ubiquitous microorganisms of the analyzed rhizosphere. *Serendipita* genus was one of the dominant genera among the common OTUs identified in our samples, which is not surprising considering its aforementioned mycorrhizal characteristics. Moreover, although not showing a particularly noticeable predominance, the *Trichoderma* genus, part of the Ascomycota phylum, was present in all our samples. These common soil colonizers have been thoroughly studied regarding their beneficial interaction with plants, particularly their significant impact on competing with phytopathogens and exchange of metabolites which can induce physiological changes in the plant and alter fruit quality (*Lombardi et al., 2020*; *de Sousa et al., 2020*; *Vitti et al., 2016*). Their antimicrobial and mycoparasitic characteristics have been extensively reviewed and have important biotechnological applications in agriculture for phytopathogens control (*Alfiky & Weisskopf, 2021*; *Harman et al., 2004*; *Reithner et al., 2011*). Thus, the presence of *Trichoderma* spp. is desired as they are naturally occurring phytopathogenic-competing microbes (*Kamble et al., 2021*; *Xu et al., 2022*). Apart from the previously mentioned taxa *Bryobacter*, *Subgroup_2*, *Candidatus Solibacter*, and *Acidothermus*, other bacterial genera stood out among the common OTUs, namely *IMCC26256*, *MB-A2-108*, *67-14*, all belonging to the Actinobacteria phylum and *Haliangium*, a Proteobacteria. Contrarily to the fungal communities, many more bacterial OTUs were common to all samples (20 ITS OTUs *vs* 1,152 16S rRNA OTUs), suggesting a much more stable bacterial community of the analyzed soils.

The application of microorganisms as a biotechnological alternative to climate change effects mitigation in plant growth, and the relevance of the rhizosphere have been the subject of extensive study and review (*Alshaal et al., 2017*; *Drigo, Kowalchuk & van Veen, 2008*; *Hakim et al., 2021*; *Naamala & Smith, 2020*; *Rodriguez & Durán, 2020*). Following this demand, this study also aimed to isolate naturally occurring microorganisms with

plant growth-promoting potential. Even though assessment of plant growth-promoting activity in blueberry plants is scarce, several of the identified fungi and bacteria belong to taxa that have been previously identified as potential PGPM and its use as biofertilizers have been proposed (*Casarrubia et al., 2020*; *de Silva et al., 2000*; *Ważny et al., 2022*). Moreover, some of them have also been revealed to enhance tolerance to heavy metals, salinity, and water stress, particularly, fungi from the *Phomopsis*, *Oidiodendron*, *Trametes*, *Trichoderma*, and *Diaporthe* genera (*da Silva Santos et al., 2022*; *Lombardi et al., 2020*; *Malik et al., 2020*; *Ważny et al., 2022*; *Xie et al., 2019b*) and bacteria from the *Bacillus*, *Enterobacter*, *Lysinibacillus*, *Micrococcus*, *Pantoea*, *Paenibacillus*, *Priestia*, *Pseudomonas*, *Acinetobacter*, *Chryseobacterium*, *Citrobacter*, *Klebsiella*, *Leclercia*, *Leifsonia*, *Staphylococcus* and *Stenotrophomonas* genera (*Afrasayab, Faisal & Hasnain, 2010*; *Ajmal et al., 2022*; *Alexander et al., 2019*; *Chhetri et al., 2022*; *Dubey et al., 2021*; *Gupta et al., 2019*; *Khan et al., 2020*; *Kour et al., 2020*; *Nascimento et al., 2020*; *Naureen et al., 2017*; *Nordstedt et al., 2021*; *Rana et al., 2020*; *Sansinenea, 2019*; *Snak et al., 2021*). From the species of fungi identified, *Mucor moelleri*, *Cunninghamella elegans*, and some species of *Trochoderma*, such as *T. harzianum*, *T. atroviride*, and *T. asperellum* have been particularly studied for their plant growth-promoting (PGP) properties (*Freitas et al., 2019*; *Hernandez et al., 2023*; *Nartey et al., 2022*; *Rao et al., 2022*; *Yu et al., 2021*). Moreover, the bacterial species *Lelliottia amigena* and *Pristia megaterium* have also been previously reported to promote plant growth (*Elakhdar, El-Akhdar & Abo-Koura, 2020*; *Sharma et al., 2022*). The isolation rate of *Trichoderma* spp. and *Bacillus* spp. stood out from the remaining taxa pointing towards a promising scale-up of biomass production of these microorganisms for in-field application. The presence of phytopathogenic agents was also identified by the presence of *Fusarium oxysposrum*, *Rosellinia necatrix*, *Setophoma terrestris*, and *Diaporthe columnaris*, which also raises the threat to the efficacy of biofertilizers and demand for the formulation of biocontrol products to combat these disease-causing agents (*Moya-Elizondo et al., 2019*; *Sawant, Song & Seo, 2022*; *Sayago et al., 2020*). Overall, the isolation of these microorganisms and their PGP potential described in the literature opens doors for further assessment of their PGP properties in *V. corymbosum* and understanding their potential for biotechnological application to reduce the impacts of climate change in the production of blueberry.

## CONCLUSIONS

Characterizing the root fungal and bacterial communities is of great interest to producers as tight links are established and could provide answers to the demand for new biotechnological alternatives to climate change consequences in plants, such as bioformulations that promote the colonization of the rhizosphere with PGPMs and antagonists of phytopathogens. In this study, the microbial structure of *V. corymbosum* 'Brigitta' rhizosphere was unveiled, disclosing a predominance of common soil colonizers among fungi and bacteria, which expectedly reflect the physicochemical composition of blueberry plants' soils and the establishment of naturally occurring mutualistic links. Even though this study focused on a single region of the country, the data obtained here is representative of a highly worldwide used blueberry-produced group and is of great use for

further studies with different Portuguese regions. Overall, we found a native microbial community that, according to what has been previously described, is capable of establishing mycorrhizal relationships and that confer beneficial physiological traits to the host plant. In order to provide applied solutions to mitigate climate change consequences in blueberry plants, we isolated and identified a plethora of naturally-occurring microorganisms that have previously been reported to have PGP activity and confer tolerance to hydric stress, a serious climate change threat. Further research should focus on testing the PGP and antagonistic potential of this collection of microorganisms in *V. corymbosum* and assess their potential to alleviate symptoms of climate change, such as drought, carbon accumulation andnutrient deficiency in blueberry plants, in order to increase their resilience against unfavorable climatic conditions and pathogenic agents.

### Funding

This work was supported by COMPETE 2020, as part of the project STOP.SUZUKII (POCI-01-0247-FEDER-047034), and by Ministério da Ciência e Tecnologia—Fundacão para a Ciência e a Tecnologia through the project financed with reference UIDB/50006/2020|UIDP/50006/2020. The funders had no role in study design, data collection and analysis, decision to publish, or preparation of the manuscript.

### Grant Disclosures

The following grant information was disclosed by the authors:
COMPETE 2020.
STOP.SUZUKII: POCI-01-0247-FEDER-047034.
Ministério da Ciência e Tecnologia—Fundacão para a Ciência e a Tecnologia: UIDB/50006/2020|UIDP/50006/2020.

### Competing Interests

The authors declare that they have no competing interests.

### Author Contributions

- Anicia Gomes conceived and designed the experiments, performed the experiments, analyzed the data, prepared figures and/or tables, authored or reviewed drafts of the article, and approved the final draft.
- Rodrigo Narciso analyzed the data, prepared figures and/or tables, authored or reviewed drafts of the article, and approved the final draft.
- Laura Regalado analyzed the data, prepared figures and/or tables, authored or reviewed drafts of the article, and approved the final draft.
- Margarida Cardeano Pinheiro performed the experiments, prepared figures and/or tables, and approved the final draft.
- Filipa Barros performed the experiments, prepared figures and/or tables, and approved the final draft.

- Sara Sario analyzed the data, prepared figures and/or tables, authored or reviewed drafts of the article, and approved the final draft.
- Conceição Santos conceived and designed the experiments, authored or reviewed drafts of the article, and approved the final draft.
- Rafael J. Mendes conceived and designed the experiments, performed the experiments, analyzed the data, prepared figures and/or tables, authored or reviewed drafts of the article, and approved the final draft.

## Field Study Permissions

The following information was supplied relating to field study approvals (*i.e.*, approving body and any reference numbers):

Field sampling was approved by Enxertada Indie Farmers under the project STOP. SUZUKII.

## Data Availability

The sequences of each sample are available at NCBI BioProject and BioSample: PRJNA940484; SAMN33569924, SAMN33569925, SAMN33569926, SAMN33569927, SAMN33569928, SAMN33569929, SAMN33569930, SAMN33569931, SAMN33569932, SAMN33569933, SAMN33569934, SAMN33569935, SAMN33569936, SAMN33569937, SAMN33569938, SAMN33569939, SAMN33569940, SAMN33569941, SAMN33569942, SAMN33569943.

## Supplemental Information

Supplemental information for this article can be found online at http://dx.doi.org/10.7717/peerj.15525#supplemental-information.

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
