# Peer review of "Disclosing the native blueberry rhizosphere community in Portugal—an integrated metagenomic and isolation approach"

_PeerJ, doi:10.7717/peerj.15525_

## Round 0.1 · original submission · Major Revisions

Dear Dr. Mendes,

Thank you for your submission to PeerJ. It is my opinion as the Academic Editor for your article - Disclosing the native blueberry rhizosphere community in Portugal - An integrated metagenomic and isolation approach - that it requires a number of Major Revisions.

We look forward to receiving your revised manuscript soon.

Reviewer 1 ·

Basic reporting

no comment

Experimental design

no comment

Validity of the findings

no comment

Additional comments

A metagenomic approach of the local fungal and bacterial community of the rhizosphere of Vaccinium corymbosum plants, and the isolation of both epiphytic and endophytic microorganisms was performed in order to disclose putative beneficial native organisms.
 
From my point of view, the work need to be major revision and review again after major revision. There are some questions should be answered.

Line 106 ”Blueberry plants, namely, Vaccinium corymbosum Brigittaí variety, grown under appropriate”
Why do you choose this blueberry variety,
Is this the most cultivated variety? Or?

Line 112-113”Ten V. corymbosum plants were randomly chosen and one soil sample per plant was collected (n= 10) (R1 to R10), to increase the representative fungal microorganisms populations at the site. ”
Why ten soil samples could increase the the representative fungal microorganisms?
And, there is an import question, do you think only ten soil samples could representative the native blueberry rhizosphere community in Portugal?

There are no results for control soil samples in the article, please add the CK results.

Line 122-124”One segment was placed directly on potato dextrose agar (PDA) medium (27 g of potato dextrose broth; 1.5% Agar; distilled water up to 1 L) supplemented with 50 mM kanamycin, to isolate epiphytic fungi”
Why do you use medium with kanamycin to isolate epiphytic fungi? Do you have some reference about that, please show it.

The 2.7. part in materials and Methods should be put after 2.2. part. OR, All culturable methods are placed after non-culturable methods in order to conform to the sequence of results described.

Annotated reviews are not available for download in order to protect the identity of reviewers who chose to remain anonymous.

Reviewer 2 ·

Basic reporting

The manuscript of Gomes et al. describes the native blueberry rhizosphere community in Portugal using an integrated metagenomic and isolation approach.
Overall, the manuscript requires professional English editing service to check the English language, spelling and grammatical errors, and sentence construction. Example is spelling error in line 17, the usage of too many ‘and’ in line 204, and many more.
In addition, the organization of the manuscript requires major reconstruction. For examples, the results section did not correspond to the methodology section. I.e., the authors described the “Metagenomic data analysis” in section 2.5, but there’s no similar section in the results. I would suggest the authors to describe their results based on the methodology section in order to create a cohesive flow between these sections. Figures were disorganized as shown in line 204-205 whereby figure 1 belongs to the 100 genera but it was mentioned latter in the sentence.

Experimental design

A lot of detail information in the methodology and results section were missing. Although some of the data were compelling, I believe that the authors need to rewrite the whole manuscript to efficiently deliver the information (see additional comments).

Validity of the findings

Why did the soil and plant samples were chosen differently? Can the author explain how they sampled the root and consider them as duplicate when the roots originated from the same plant?

Raw metagenomic data was not made available public repository.

Additional comments

1. Line 50: The citation should be in a specific order (i.e., either ascending or descending publication year)
2. Line 164-168: The method is not clear and needs clarification. I.e., the authors first mentioned that they used “the top 10 highest abundant fungal and bacterial genus” but then used 100 and 10 genera for the construction of evolutionary and phylogenetic tree, respectively. What were the rationale behind this number selection?
3. Section 2.3: Include the number of soil samples, fungal and bacterial isolates that were used for DNA extraction.
4. Section 2.4: Specify the samples used in the heading. I.e., ITS and 16S amplicon metagenomic sequencing of the V. corymbosum rhizosphere.
5. Include the number of libraries that were constructed.
6. Line 174 (section 2.6) needs to be rewritten. The primer sequences can be included in a table instead of putting them in the main text.
7. Section 3.1 and 3.2 can be combined under “Metagenomic data analysis” heading.
8. Table 3 is not mentioned anywhere in the main text.

---

## Round 0.2 · accepted · Accept

The authors have addressed the questions and concerns raised by the reviewers well. I am satisfied with the current version of the manuscript and therefore decide to proceed in accepting the publication of the manuscript.

Reviewer 1 ·

Basic reporting

no comment

Experimental design

no comment

Validity of the findings

no comment

Additional comments

In this study, a metagenomic approach of the local fungal and bacterial community of the rhizosphere of Vaccinium corymbosum plants was performed. At the same time, both epiphytic and endophytic microorganisms were isolated in order to disclose putative beneficial native organisms.
From my point of view, the work have been done lots of revision according to the suggestion. And I think the work could be accepted.